# Extracellularly Released Calreticulin Induced by Endoplasmic Reticulum Stress Impairs Syncytialization of Cytotrophoblast Model BeWo Cells

**DOI:** 10.3390/cells10061305

**Published:** 2021-05-24

**Authors:** Naoyuki Iwahashi, Midori Ikezaki, Kazuchika Nishitsuji, Madoka Yamamoto, Ibu Matsuzaki, Naoki Kato, Naoyuki Takaoka, Mana Taniguchi, Shin-ichi Murata, Kazuhiko Ino, Yoshito Ihara

**Affiliations:** 1Department of Obstetrics and Gynecology, Wakayama Medical University School of Medicine, 811-1 Kimiidera, Wakayama 641-8509, Japan; naoyuki@wakayama-med.ac.jp (N.I.); madoka-y@wakayama-med.ac.jp (M.Y.); kazuino@wakayama-med.ac.jp (K.I.); 2Department of Biochemistry, Wakayama Medical University School of Medicine, 811-1 Kimiidera, Wakayama 641-8509, Japan; ikezaki@wakayama-med.ac.jp (M.I.); hwhbg038@yahoo.co.jp (N.K.); g.200swimmer@gmail.com (N.T.); qscbbv27476@gmail.com (M.T.); 3Department of Human Pathology, Wakayama Medical University School of Medicine, 811-1 Kimiidera, Wakayama 641-8509, Japan; m_ibu@wakayama-med.ac.jp (I.M.); smurata@wakayama-med.ac.jp (S.-i.M.)

**Keywords:** calreticulin, syncytialization, trophoblast, preeclampsia, endoplasmic reticulum stress

## Abstract

The pregnancy-specific syndrome preeclampsia is a major cause of maternal mortality throughout the world. The initial insult resulting in the development of preeclampsia is inadequate trophoblast invasion, which may lead to reduced maternal perfusion of the placenta and placental dysfunction, such as insufficient trophoblast syncytialization. Endoplasmic reticulum (ER) stress has been implicated in the pathology of preeclampsia and serves as the major risk factor. Our previous studies suggested critical roles of calreticulin (CRT), which is an ER-resident stress response protein, in extravillous trophoblast invasion and cytotrophoblast syncytialization. Here, we studied the mechanism by which ER stress exposes the placenta to the risk of preeclampsia. We found that CRT was upregulated in the serum samples, but not in the placental specimens, from preeclamptic women. By using BeWo cells, an established model of cytotrophoblasts that syncytialize in the presence of forskolin, we demonstrated that thapsigargin-induced ER stress caused extracellular release of CRT from BeWo cells and that the extracellular CRT suppressed forskolin-induced release of β-human chorionic gonadotropin and altered subcellular localization of E-cadherin, which is a key adhesion molecule associated with syncytialization. Our results together provide evidence that induction of ER stress leads to extracellular CRT release, which may contribute to placental dysfunction by suppressing cytotrophoblast syncytialization.

## 1. Introduction

The placenta in humans is critical for embryonic development and pregnancy maintenance. The placenta and associated extraembryonic membranes are extraembryonic structures of the conceptus and form from the zygote at the initial stage of pregnancy. The trophectoderm, the precursor of all trophoblast cells, is the first epithelium that appears during mammalian embryogenesis; it is a polarized single-cell layer that makes up the blastocyst wall. During embryogenesis, the trophectoderm forms the cytotrophoblasts (CTBs), or epithelial “stem cells” of the placenta, that differentiate into two major placental cell types: extravillous trophoblasts (EVTs) and syncytiotrophoblasts (STBs) [1]. The EVTs are involved in uterine artery remodeling, which is crucial for perfusion of the placenta with maternal blood; CTBs undergo syncytialization to form multinucleated STBs that are essential for nutrient and gas exchange at the maternal–fetal interface and for hormone synthesis to support the pregnancy [2,3,4]. Dysfunction of the CTB syncytialization may result in pregnancy-related pathologies such as preeclampsia, which is characterized by hypertension, proteinuria, and edema and is a major cause of maternal death [5,6,7].

The detailed molecular mechanisms of the development and progression of preeclampsia are not fully understood. Some data have suggested the involvement of endoplasmic reticulum (ER) stress in the pathogenesis of human pregnancy complications including preeclampsia [8,9,10,11]. Although inadequate placentation has been suggested to result in the induction of ER stress [12], how ER stress contributes to preeclampsia development is unknown. We previously showed that calreticulin (CRT), an ER-resident molecular chaperone that is ubiquitously expressed throughout the body including the placenta [13,14], is necessary for both adequate invasion of EVTs and syncytialization of CTBs [15,16]. Certain studies reported increases in CRT mRNA and protein levels in maternal blood and placentas from preeclamptic patients [12,13]. Although CRT is a classical ER-resident chaperone, it has been observed outside the ER, cell surface, and extracellular compartments, and it regulates various biological processes such as uptake by dendritic cells or phagocytosis of CRT-expressing cancer cells and apoptotic cells [17,18], cell migration, and cell proliferation [19,20,21,22,23]. Extracellular release of CRT is an unusual event, however, and because CRT is a stress response protein, ER stress may be involved in the extracellular release of CRT [24]. Adding exogenous Escherichia coli-expressed human CRT to the culture medium for the HTR8/SVneo human trophoblast cell line reportedly reduced cell migration [25]. These lines of evidence prompted us to study whether ER stress induces extracellular release of CRT from CTBs and whether this extracellular CRT affects the functions of CTBs, such as syncytialization.

We found here that serum CRT levels were significantly higher in preeclamptic patients than in women with normal pregnancies. We used thapsigargin as an ER stress inducer and human choriocarcinoma BeWo cells as a CTB model, and we discovered that extracellular CRT was released as a result of ER stress in BeWo cells. The extracellularly released CRT reduced forskolin-stimulated release of β-human chorionic gonadotropin (β-hCG) and altered the cell surface localization of E-cadherin, which is a cell−cell adhesion protein and is thus critical for CTB syncytialization [26]. On the basis of our results, we thus propose a novel non-ER function of CRT that may contribute to dysfunctional placentation and the development of preeclampsia.

## 2. Materials and Methods

### 2.1. Materials

The anti-CRT antibody, anti-calnexin antibody, anti-immunoglobulin-binding protein (BiP) antibody, and anti-ER-resident protein 57 (ERp57) antibody were obtained from Stressgen Biotechnologies (San Diego, CA, USA). The anti-E-cadherin antibody, anti-β-catenin antibody, anti-glyceraldehyde-3-phosphate dehydrogenase (GAPDH) antibody, and anti-Golgi matrix protein of 130 kDa (GM130) antibody were from BD Biosciences (San Jose, CA, USA), Sigma-Aldrich (St. Louis, MO, USA), Merck Millipore (Burlington, MA, USA), and Abcam (Cambridge, UK), respectively.

### 2.2. Human Tissue Collection and Sample Preparation

We obtained informed consent from individual patients for the use of the placental specimens. Third-trimester samples of human placenta were collected during performance of cesarean sections before the onset of labor. Preeclampsia was defined as having a blood pressure reading of ≥140/90 mmHg after 20 weeks of gestation plus proteinuria (≥300 mg of protein/24 h). The eligibility of the preeclampsia cases was determined on the basis of the diagnostic criteria of the International Society for the Study of Hypertension in Pregnancy. Cases involving intrauterine growth retardation, multiple pregnancies, fetal chromosomal abnormalities, or fetal anomalies were excluded. For immunoblot analysis, we collected specimens of placentas from patients with preeclampsia (n = 6) and women with gestational age-matched normal pregnancies (n = 6), as well as serum samples from healthy nonpregnant women (n = 6), women with gestational age-matched normal pregnancies (n = 6), and patients with preeclampsia (n = 6). The patients used in the immunoblot analysis are included in Table 1. Placental specimens and serum samples were immediately frozen in liquid nitrogen and stored at −80 °C before being used for analysis. This study was approved by the Ethics Committee of Wakayama Medical University.

### 2.3. Immunohistochemistry

Paraffin-embedded blocks of placental tissue samples (patients with preeclampsia: n = 18; gestational age-matched controls: n = 22) were cut into 3-μm-thick sections, deparaffinized, and rehydrated. We then blocked endogenous peroxidase activity with 0.3% hydrogen peroxide in methanol. For antigen retrieval, we boiled the slides in 1 mM ethylenediaminetetraacetic acid (pH 8.0) in a pressure cooker for 10 min. We incubated the sections with an anti-CRT antibody (Abcam) at a dilution of 1:5000. We detected signals by using the simple stain MAX-PO reagent (Nichirei Biosciences, Tokyo, Japan) and 3,3-diaminobenzidine as the substrate. We used Mayer’s hematoxylin solution for counterstaining and scored the staining intensity as 0 = none, 1 = weak, 2 = moderate, and 3 = strong staining. We graded the percentage of positive cells as 0: no positive cells, 1: <30%, 2: 30–80%, or 3: >80% and then used the following equation to calculate the immunoreactive score (IRS): IRS = staining intensity × percentage of positive cells, according to previous reports [16,27].

### 2.4. Cell Culture

Human choriocarcinoma BeWo cells, which are commonly used as a model of trophoblast differentiation and syncytialization [5,28]), were purchased from the American Type Culture Collection (Manassas, VA, USA) and were authenticated by the JCRB Cell Bank (National Institute of Biomedical Innovation-Japan, Report No. KBN0410). BeWo cells were grown in Roswell Park Memorial Institute RPMI 1640 medium (Wako Pure Chemicals, Osaka, Japan) supplemented with 10% fetal calf serum (BioWest, Nuaillé, France) and penicillin, streptomycin, and amphotericin B (Life Technologies, Carlsbad, CA, USA) at 37 °C in 5% CO_2_ and 95% air.

### 2.5. Reverse Transcription-Polymerase Chain Reaction (RT-PCR)

Total RNA was collected and reverse-transcribed by using PrimeScript One Step RT-PCR Kit Ver. 2 (TaKaRa Biomedicals, Shiga, Japan). Polymerase chain reactions were performed by using the following cycling parameters: for human X-box-binding protein-1 (XBP-1): 25 cycles at 94 °C for 0.5 min, at 65 °C for 0.5 min, and at 72 °C for 1 min; for human GAPDH: 20 cycles at 94 °C for 0.5 min, at 56 °C for 0.5 min, and at 72 °C for 1 min. The primer sequences were as follows: for humanXBP-1: forward 5′-GCT GAA GAG GAG GCG GAA G-3′, reverse 5′-GTC CAG AAT GCC CAA CAG G-3′; for human GAPDH: forward 5′-GGA TTT GGT CGT ATT GGG CG-3′, reverse 5′-CAG TAG AGG CAG GGA TGA TG-3′. We analyzed the transcripts by using 8% polyacrylamide gel electrophoresis for XBP-1 and 1.5% agarose gel electrophoresis for GAPDH followed by ethidium bromide staining. GAPDH served as a reference RNA.

### 2.6. Immunodepletion

BeWo cells were treated with thapsigargin (25–500 nM, Wako Pure Chemicals) for 1 h at 37 °C, after which cells were washed and culture media were replaced with fresh Opti-MEM (Thermo Fisher Scientific, Waltham, MA, USA). After a 24-h culture at 37 °C, we collected the conditioned medium [thapsigargin-conditioned medium (Tg-CM)]. For immunodepletion of CRT from Tg-CM, CRT was immunoprecipitated by using anti-CRT antibody that was conjugated with Dynabeads Protein G (Thermo Fisher Scientific). We used the supernatants as CRT-depleted Tg-CM. The absence of CRT in supernatants was confirmed by immunoblot analysis (data not shown). Purified rabbit IgG (Thermo Fisher Scientific) served as an isotype control.

### 2.7. Immunoblot Analysis

We used a Teflon homogenizer (AS ONE Corporation, Osaka, Japan) to homogenize 500-mg samples of human placental tissues in 0.4 mL of RIPA buffer (Sigma-Aldrich) containing the protease inhibitors 4 mM Pefabloc, 1 μM pepstatin, 1 μM leupeptin, and 200 μM phenylmethylsulfonyl fluoride (Roche, Basel, Switzerland). Samples were then centrifuged at 18,000× *g* for 10 min at 4 °C, and supernatants were collected and subjected to immunoblot analysis as described below. To induce ER stress, we treated BeWo cells with thapsigargin (25–500 nM) for 1 h at 37 °C, and we analyzed the induction of ER stress by using immunoblotting with anti-phosphorylated or anti-pan-eukaryotic translation initiation factor 2α antibodies (Cell Signaling Technology, Danvers, MA, USA). For cell-based assays, we treated BeWo cells with Tg-CM or CRT-depleted Tg-CM for 24 h at 37 °C in the presence of forskolin (80 μM, Sigma-Aldrich). Cells were then lysed in lysis buffer A (10 mM Tris-HCl (pH 7.5), 150 mM NaCl, 1% Nonidet P-40, and protease inhibitors). Cell lysates were sonicated intermittently on ice for 15 min and then centrifuged at 10,000× *g* for 10 min at 4 °C. Supernatants were separated by using 10% sodium dodecyl sulfate-polyacrylamide gel electrophoresis and were then transferred to polyvinylidene difluoride membranes (Immobilon-P; Merck Millipore). For analysis of serum samples, each sample was diluted with tris-buffered saline (pH 7.6) to obtain the final protein concentration of 10 µg/µL, and 100 µg proteins were applied to each well of 10% sodium dodecyl sulfate-polyacrylamide gels. Protein concentrations of nondiluted serum samples are shown in Appendix A. Membranes were blocked with the EzBlock Chemi blocking solution (ATTO, Tokyo, Japan) and were incubated with primary antibodies at 4 °C overnight followed by incubation with peroxidase-conjugated secondary antibodies (Cell Signaling Technology). Signals were detected by using the Immobilon Western Chemiluminescent Horseradish Peroxidase substrate (Merck Millipore) and were densitometrically quantified by using ImageJ version 1.50b (National Institutes of Health, Bethesda, MD, USA). GAPDH served as a loading control.

### 2.8. Cytotoxicity of Thapsigargin Treatment

We analyzed cytotoxicity by means of the lactate dehydrogenase (LDH) release assay. We treated BeWo cells with thapsigargin (25–500 nM) for 1 h at 37 °C and measured LDH release into culture media by using the MTX-LDH kit (Kyokuto Pharmaceutical, Tokyo, Japan).

### 2.9. Immunocytochemistry and Determination of Fusion Index

BeWo cells were grown on coverslips cultured for 24 h, after which cells were treated with Tg-CM or CRT-depleted Tg-CM for 24 h at 37 °C in the presence of forskolin (80 μM). Cells were then fixed with 4% paraformaldehyde in PBS for 20 min at room temperature and permeabilized with 0.5% Triton X-100 in PBS for 10 min. After the cells were washed 3 times with PBS, they were blocked with 3% bovine serum albumin in PBS for 30 min and incubated with primary antibodies followed by Alexa 488- or Alexa 555-conjugated secondary antibodies (Thermo Fisher Scientific). Stained specimens were mounted with Vectashield mounting medium containing DAPI (Vector Laboratories, Burlingame, CA, USA) and examined with a LSM700 laser scanning confocal microscopy and the LSM software ZEN 2012 (Carl Zeiss, Jena, Germany). For determination of fusion indexes, numbers of nuclei in STB-like fused cells and the total numbers of nuclei were counted in 8 to 10 randomly selected microscopic fields in each E-cadherin-stained specimen. Fusion indexes were calculated as the percentages of the total number of nuclei in the STB-like fused cells to the total number of nuclei in the microscopic fields.

### 2.10. Statistical Analysis

Data were analyzed via one-way ANOVA with Bonferroni’s multiple comparisons test, or the Mann−Whitney U-test by means of Prism software (Version 7.04, GraphPad Software, San Diego, CA, USA). *p* values of <0.05 were said to be significant.

## 3. Results

### 3.1. Upregulation of Serum CRT Levels in Patients with Preeclampsia

Table 1 summarizes the clinical characteristics of the patients in the present study. We first used Western blotting to determine the CRT protein levels in the samples of placenta (Figure 1a) and serum (Figure 1b) collected from nonpregnant women (n = 6), women with gestational age-matched normal pregnancies (n = 6), and preeclamptic patients (n = 6), who were included in Table 1. We successfully detected the 55-kDa CRT protein in placental and serum samples. We used cell lysates obtained from BeWo cells as the reference for the CRT protein band [16]. We noted no difference in the CRT levels in placentas from women with normal pregnancies and women with preeclampsia (Figure 1c), but the serum CRT levels were approximately 50% lower in nonpregnant women and 35% higher in preeclamptic women than the levels in women with normal pregnancies (Figure 1d). We also determined the CRT levels in the third-trimester placentas of the patients shown in Table 1 by using immunohistochemical analysis (normal placentas (n = 22) and preeclamptic placentas (n = 18); Figure 1e). In agreement with our previous report [16], the major site of CRT expression was the STB cytoplasm in both normal and preeclamptic placentas (Figure 1e). We also detected CRT in endothelial cells and villous macrophages (Hofbauer cells) in the villous cores, EVTs in the decidua and chorionic membrane, and connective tissue cells in the amnion layer of the external membranes in third-trimester placentas. We noted no difference in IRSs for normal placentas and preeclamptic placentas, a finding that confirmed the immunoblot analysis (Figure 1e).

Data are expressed as means ± SD; ns, not significant.

### 3.2. Thapsigargin-Induced ER Stress in BeWo Cells

Because ER stress in placentas has been implicated in the etiology and pathology of human pregnancy complications, including preeclampsia [29,30], we hypothesized that the induction of ER stress would stimulate placental cells to release CRT, which may contribute to increased serum CRT levels in preeclamptic patients (Figure 1b,d). We utilized human choriocarcinoma BeWo cells, which are commonly used as a model of trophoblast differentiation and syncytialization [5,28], and thapsigargin, which is a very popular ER stress inducer and depletes the calcium ion store in the ER, thereby causing ER stress [31]. We treated BeWo cells with thapsigargin (25–500 nM) for 1 h at 37 °C, after which we replaced the culture media with fresh media and continued the cell culture for 23 h at 37 °C. We then harvested the cells, induced ER stress, and determined the cytotoxicity of the thapsigargin treatment. The XBP-1 splicing assay [32] confirmed the thapsigargin concentration-dependent induction of ER stress (Figure 2a). Immunoblot analysis of phosphorylated eukaryotic initiation factor-2α (eIF2α), another marker of ER stress induction [33], also indicated thapsigargin concentration-dependent induction of ER stress (Figure 2b). We did not observe extracellular release of LDH after thapsigargin treatment, which excluded the possibility of plasma membrane damage (Figure 2c).

### 3.3. Extracellular Release of CRT after Thapsigargin-Induced ER Stress in BeWo Cells

We next investigated the effect of thapsigargin-induced ER stress on the release of CRT from BeWo cells. We treated BeWo cells with thapsigargin (25–500 nM) for 1 h at 37 °C, after which culture media were replaced with fresh media and cells were cultured for 23 h at 37 °C. We then harvested the conditioned media and analyzed the CRT levels in the conditioned media. Immunoblots with an anti-CRT antibody showed thapsigargin concentration-dependent extracellular release of CRT from BeWo cells (Figure 3a). Because we did not observe extracellular release of BiP from thapsigargin-treated BeWo cells (Figure 3a), we concluded that ER stress did not induce extracellular release of all ER-resident proteins, that is, this phenomenon was specific to CRT. We also used immunoblot analysis to determine the expression of CRT and other ER-resident proteins including BiP, calnexin, and ERp57 in BeWo cells treated with 100 nM thapsigargin. We detected no alterations in the expression of these proteins including CRT after thapsigargin treatment (Figure 3b). We previously reported that genetic ablation of CRT expression resulted in reduced cell surface localization of E-cadherin and subsequent fusion of BeWo cells [16]. In our study here, we detected two bands that corresponded to E-cadherin, similar to bands for epithelial cells whose trafficking of E-cadherin was inhibited by cellular stress (Figure 3b) [34]. As expected, our immunofluorescence analysis showing enhanced and condensed colocalization of E-cadherin and the cis-Golgi marker GM130 confirmed thapsigargin-induced altered E-cadherin subcellular localization (Figure 3c).

### 3.4. Forskolin-Induced Syncytialization of BeWo Cells Inhibited by Extracellular CRT

We also studied the effects of extracellular CRT on the functions of BeWo cells. Because BeWo cells secrete β-hCG, which is mainly produced by placental STBs and promotes syncytialization in an autocrine−paracrine manner [35,36], we investigated whether extracellular CRT would affect forskolin-induced β-hCG secretion in BeWo cells. Because we successfully induced ER stress and extracellular release of CRT by 100 nM thapsigargin treatment (Figure 2a,b and Figure 3a), we used 100 nM thapsigargin treatment for further experiments. We treated BeWo cells with 100 nM thapsigargin for 1 h at 37 °C, after which culture media were replaced with fresh media and cells were cultured for 23 h at 37 °C. We then collected Tg-CM samples and treated fresh BeWo cells with Tg-CM in the presence of 80 μM forskolin for 24 h at 37 °C. We successfully obtained CRT-containing Tg-CM by using thapsigargin treatment (Figure 4a). Because we solubilized thapsigargin in dimethyl sulfoxide (DMSO), we prepared DMSO-treated conditioned medium (DMSO-CM) as the control. Figure 4b,c show that treatment with Tg-CM markedly suppressed forskolin-induced β-hCG secretion and cell fusion in BeWo cells. Tg-CM-treated BeWo cells showed two E-cadherin immunoreactive bands that were similar to those of thapsigargin-treated cells (Figure 3b and Figure 4b), which suggests that extracellular CRT may affect subcellular localization of E-cadherin. Immunofluorescence analysis demonstrated enhanced and condensed colocalization of E-cadherin and the cis-Golgi marker, which confirmed the unfavorable effect of extracellular CRT on syncytialization (Figure 4d).

### 3.5. Elimination of Harmful Effects of Tg-CM on BeWo Cells by Means of CRT Immunodepletion

Lastly, we immunodepleted CRT proteins in Tg-CM by using an anti-CRT antibody and Dynabeads that were conjugated with protein G (CRT-depleted CM). By using immunoblot analysis, we found that CRT-depleted CM lacked the suppressive effect of Tg-CM on forskolin-induced β-hCG secretion and cell fusion, which strongly supported the adverse effect of extracellular CRT on syncytialization (Figure 5a,b). In addition, enhanced and condensed colocalization patterns of E-cadherin and GM130 disappeared in CRT-depleted CM-treated BeWo cells (Figure 5c). Together, our results demonstrated the detrimental effects of ER stress-induced extracellular release of CRT on CTB functions such as β-hCG secretion and syncytialization, thus leading to the ER stress-induced development of preeclampsia.

## 4. Discussion

CRT is an ER-resident molecular chaperone that is ubiquitously expressed throughout the body. In addition, non-ER functions of CRT have been implicated in many physiological and pathological processes, including wound healing, immunity, cell development, proliferation, differentiation, malignancy, and tumor progression [14,37,38]. In the ER, CRT plays important roles not only in the proper folding of newly synthesized proteins and glycoproteins but also in homeostatic regulation of calcium levels in the cytosol and ER [14,39]. The human placenta is one site with high CRT expression during pregnancy [40,41], which suggests a role of CRT in placentation and maintenance of pregnancy. We previously showed that CRT expression by placental cells was required for trophoblast invasion and trophoblast syncytialization [15,16], which supported the importance of CRT in pregnancy. CRT may be released from cells under certain conditions, such as injury or death [42]. Although CRT at the cell surface reportedly served as a marker of phagocytic clearance [17,18], the functions of extracellular CRT in placentas have remained unclear. In our study here, we found that CRT was released from cells because of ER stress and that extracellular CRT possessed certain biological functions such as reduction of forskolin-induced β-hCG secretion and modification of membrane localization of E-cadherin. Our results showing that extracellularly released CRT inhibited forskolin-induced syncytialization will aid our understanding of the molecular functions of CRT beyond that of an ER chaperone. A previous study reported that a tumor-homing peptide recognized CRT at the placental membrane [6], which suggested a potential non-ER function of CRT in the targeted delivery of materials to the placenta. Our study here, however, is the first to report a non-ER extracellular function of CRT—involvement in the development of preeclampsia. Tong et al. identified CRT as one component of trophoblast extracellular microvesicles [43]. Because extracellular vesicles such as exosomes and microvesicles are involved in cell−cell communication in that they transfer biomolecules including microRNAs and proteins [44], the role of CRT on or in extracellular vesicles warrants future investigation.

ER stress is one major risk factor for preeclampsia, as suggested by the upregulation of several ER stress markers in placentas with both intrauterine growth retardation and preeclampsia compared with placentas with intrauterine growth retardation alone [45,46]. Increased ER stress may lead to reduced proliferation and enhanced apoptosis of trophoblasts [45], but the precise mechanism of how ER stress results in preeclampsia is unknown. Gu et al. reported a significant increase in blood CRT levels, but not in placental CRT levels, in women with normal pregnancies compared with nonpregnant women [13]. In the present study, blood samples from women with normal pregnancies had significantly higher CRT levels than samples from nonpregnant women, and blood samples from preeclamptic women demonstrated further marked increases in CRT levels compared to those from women with normal pregnancies. Furthermore, BeWo cells released CRT in response to ER stress, but not under general culture conditions without ER stress induction, and the extracellular CRT prevented forskolin-induced syncytialization. Thus, our results provide evidence that excess extracellular CRT may have adverse effects on placental development and cause dysfunctional placentation that may lead to preeclampsia.

The molecular mechanism by which extracellular CRT suppressed forskolin-induced β-hCG release is as yet unclear. In extracellular CRT-treated BeWo cells, we observed an E-cadherin band in addition to the mature 120 kDa form, which may correspond to the pro-E-cadherin [34], and extracellular CRT induced E-cadherin to accumulate in the GM130-positive cis-Golgi. Although the accumulation of E-cadherin in the cis-Golgi was similar to that reported in our previous study of BeWo cells whose CRT was genetically depleted by stable expression of short hairpin RNA against CRT [16], we did not observe the additional E-cadherin band in that study. Thus, it seems to be unlikely that extracellular CRT interfered with the chaperone function of CRT on E-cadherin, and our current results suggest another role of extracellular CRT in determining subcellular localization of E-cadherin. Elucidating the details of the detrimental effects of extracellular CRT is a future challenge.

The source of the increased serum CRT in preeclamptic patients is also not fully clarified. Gu et al. suggested that the placenta may be a major source of circulating CRT in pregnancy [13]. In view of the induction of ER stress in preeclamptic placentas and our results showing the extracellular release of CRT by an ER stress inducer, the placenta may indeed be the major source of increased circulating CRT in preeclamptic serum. However, our immunohistochemical and immunoblot analyses of human placental tissues demonstrated no significant differences between placental CRT protein levels in women with normal pregnancies and in patients with preeclampsia. Thus, we cannot discount the possibility that tissues and organs other than the placenta may contribute to circulating CRT levels.

## 5. Conclusions

In summary, we found increased CRT levels in samples of serum from preeclamptic patients but not in samples of placentas. The results of the cell-based assays supported the pathological roles of extracellular CRT in preeclampsia. Our findings are thus important for the prediction, prevention, and treatment of preeclampsia. We showed that ER stress, a major risk factor for preeclampsia, induced extracellular rerelease of CRT and that the extracellular CRT prevented forskolin-induced syncytialization in CTB model cells, but elucidating the precise molecular mechanisms of these data deserves additional study.

## Figures and Tables

**Figure 1 cells-10-01305-f001:**
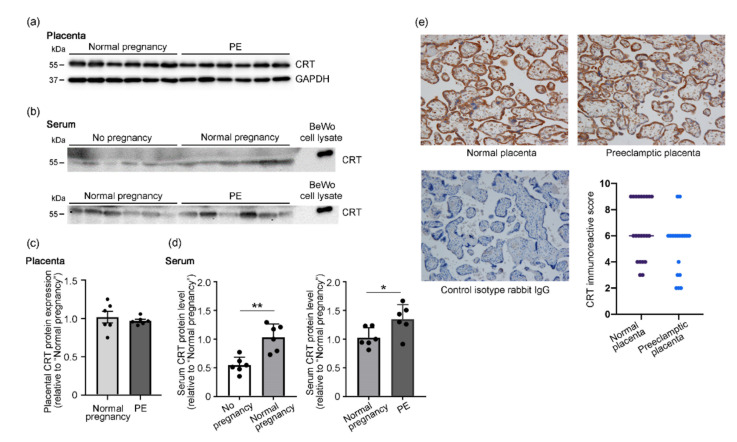
Upregulation of serum CRT levels in patients with preeclampsia. CRT protein levels in placental (**a**) and serum (**b**) samples obtained from nonpregnant women (n = 6), women with normal pregnancies (n = 6), and patients with preeclampsia (PE; n = 6) were analyzed by using Western blotting. Cell lysates that were prepared from BeWo cells under conventional culture conditions served as the reference for the CRT-immunoreactive band. GAPDH was used as the loading control in (**a**). For analysis of serum samples in (**b**), 100 µg of total proteins were loaded in each well. (**c**,**d**) Quantitative analysis of the immunoblots in (**a**) and (**b**). (**e**) Representative images of immunohistochemical analysis of CRT in normal placentas (n = 22) and preeclamptic placentas (n = 18). The graph provides CRT IRSs. Data are presented as means ± standard deviation (SD). * *p* < 0.05; ** *p* < 0.01.

**Figure 2 cells-10-01305-f002:**
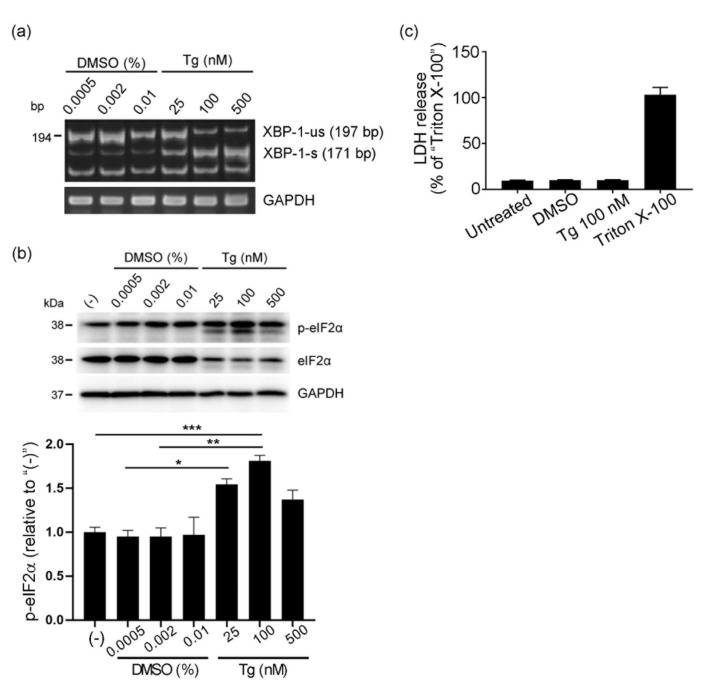
Thapsigargin-induced ER stress in BeWo cells. BeWo cells were treated with thapsigargin (Tg, 25–500 nM) or DMSO for 1 h at 37 °C. After samples were washed three times with phosphate-buffered saline (PBS), culture media were replaced with fresh Opti-MEM and cells were cultured for 24 h at 37 °C. Cells were then harvested, and induction of ER stress and cytotoxicity of thapsigargin treatment were analyzed. (**a**) mRNAs of XBP-1 and GAPDH were assessed by using RT-PCR. XBP 1-s, spliced fragments of XBP-1; XBP 1-us, unspliced fragments of XBP-1. (**b**) Expression levels of the ER stress marker phosphorylated eIF2α (p-eIF2α) were analyzed by using Western blotting. GAPDH served as the loading control. The graph shows quantification of p-eIF2α. Data represent means ± SE (n = 3). (**c**) Cytotoxicity of thapsigargin treatment (Tg) was assessed by means of the LDH release assay. Results are presented as a percentage of the values of 2% Triton X-100-treated cells, which caused the maximum release via complete membrane disruption. Data represent means ± SE (n = 5). * *p* < 0.05; ** *p* < 0.01; *** *p* < 0.001.

**Figure 3 cells-10-01305-f003:**
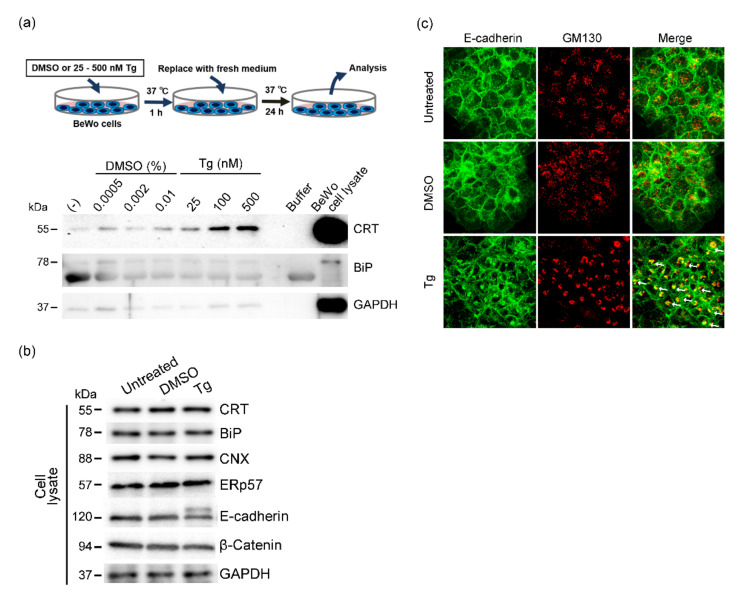
Extracellular release of CRT after thapsigargin-induced ER stress in BeWo cells. (**a**) BeWo cells were treated with thapsigargin (Tg) or DMSO for 1 h at 37 °C. After samples were washed three times with PBS, culture media were replaced with fresh Opti-MEM and cells were cultured for 24 h at 37 °C. Conditioned medium samples were then collected and CRT protein levels were analyzed by means of Western blotting. GAPDH and BiP served as markers for the cytosol and the ER, respectively. (**b**) Expression of ER-resident proteins in thapsigargin-treated (Tg, 100 nM) BeWo cells was analyzed by using Western blotting. GAPDH served as the loading control. (**c**) DMSO- or thapsigargin-treated (Tg, 100 nM) BeWo cells were stained with anti-E-cadherin and anti-GM130 antibodies. Arrows point to condensed E-cadherin signals that colocalized with GM130.

**Figure 4 cells-10-01305-f004:**
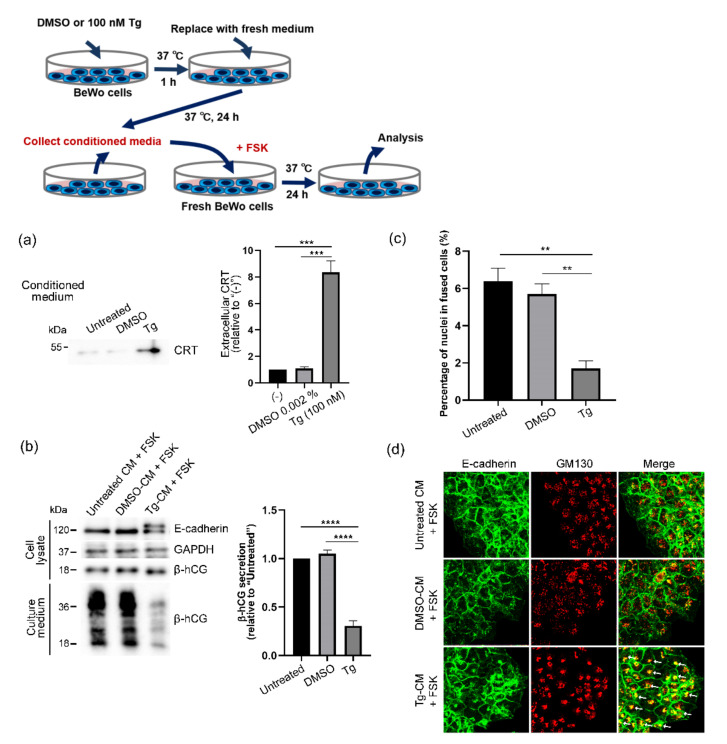
Extracellular CRT-induced inhibition of forskolin-induced syncytialization in BeWo cells. (**a**) BeWo cells were treated with thapsigargin (Tg) or DMSO for 1 h at 37 °C. After samples were washed three times with PBS, culture media were replaced with fresh Opti-MEM and cells were cultured for 24 h at 37 °C. Conditioned medium samples were then collected to obtain DMSO-CM and Tg-CM. Extracellular release of CRT by thapsigargin was confirmed by means of Western blotting. The graph shows quantification of extracellular release of CRT by Tg (100 nM)-treatment. Data represent means ± SE (n = 3). (**b**) To investigate the effects of extracellular CRT on syncytialization, fresh BeWo cells were treated with DMSO-CM or Tg-CM in the presence of forskolin (FSK, 80 µM) for 24 h at 37 °C, and expression of E-cadherin and secretion of β-hCG were analyzed by using Western blotting. GAPDH served as the loading control. The graph shows quantification of secreted β-hCG. Data represent means ± SE (n = 3). (**c**) Effects of Tg-CM treatment on syncytialization was analyzed by means of fusion indexes. Data represent means ± SE (n = 3). (**d**) Effects of Tg-CM treatment on subcellular localization of E-cadherin were analyzed by means of immunofluorescence with anti-E-cadherin and anti-GM130 antibodies. Arrows point to condensed E-cadherin signals that colocalized with GM130. ** *p* < 0.01; *** *p* < 0.001; **** *p* < 0.0001.

**Figure 5 cells-10-01305-f005:**
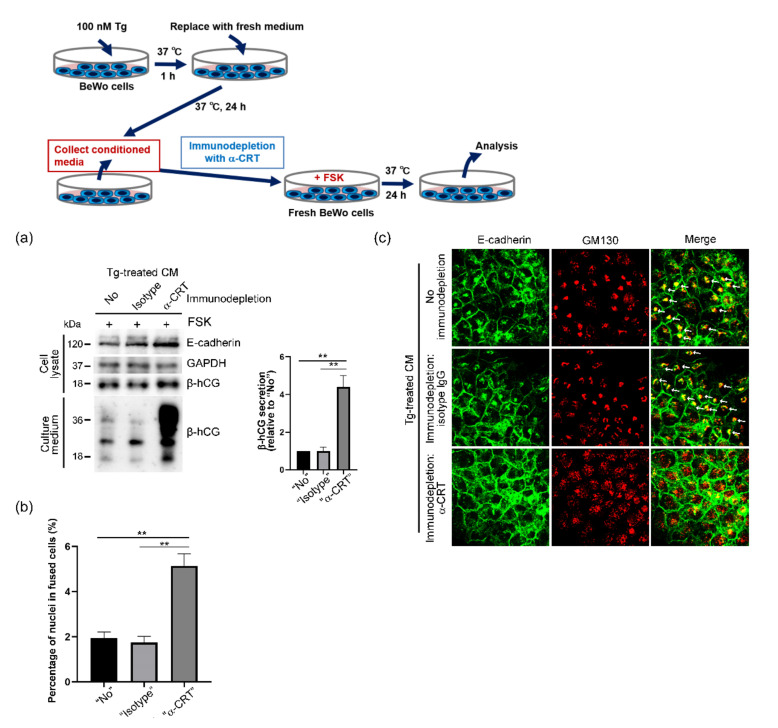
Elimination of detrimental effects of Tg-CM on BeWo cells by CRT immunodepletion. (**a**) BeWo cells were treated with thapsigargin (Tg) for 1 h at 37 °C. After samples were washed three times with PBS, culture media were replaced with fresh Opti-MEM and cells were cultured for 24 h at 37 °C. Conditioned medium samples were then collected to obtain Tg-CM. CRT in Tg-CM was immunodepleted by using an anti-CRT antibody and Dynabeads with protein G. Fresh BeWo cells were then treated with Tg-CM or CRT-depleted Tg-CM in the presence of forskolin (FSK, 80 µM) for 24 h at 37 °C, and expression of E-cadherin and secretion of β-hCG were analyzed via Western blotting. The graph shows quantification of secreted β-hCG. Data represent means ± SE (n = 3). (**b**) Effects of CRT immunodepletion from Tg-CM on syncytialization was analyzed by means of fusion indexes. Data represent means ± SE (n = 3). (**c**) Fresh BeWo cells were treated with Tg-CM or CRT-depleted Tg-CM in the presence of forskolin (80 µM) for 24 h at 37 °C, and subcellular localization of E-cadherin was analyzed by using immunofluorescence with anti-E-cadherin and anti-GM130 antibodies. Arrows point to condensed E-cadherin signals that colocalized with GM130. ** *p* < 0.01.

**Table 1 cells-10-01305-t001:** Clinical characteristics of the study population.

	Normal Pregnancy	Preeclampsia Alone	*p* Value
Characteristic	(n = 22)	(n = 18)	
Maternal age (years)	30.4 ± 5.5	32.8 ± 4.6	ns
Prepregnancy body mass index (kg/m^2^)	20.7 ± 2.6	25.5 ± 5.9	<0.001
Gestational age at delivery (weeks)	34.7 ± 4.2	34.2 ± 2.6	ns
Neonatal weight (g)	2447 ± 712	2232 ± 614	ns
Placental weight (g)	528 ± 133	512 ± 129	ns

## Data Availability

The data presented in this study are openly available in FigShare at https://doi.org/10.6084/m9.figshare.14658768.

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
