# Peer review of "Extracellularly Released Calreticulin Induced by Endoplasmic Reticulum Stress Impairs Syncytialization of Cytotrophoblast Model BeWo Cells"

_cells, 2021, doi:10.3390/cells10061305_

Round 1

Reviewer 1 Report

This manuscript outlines the effect of extracellular calreticulin (CRT),  an ER resident chaperone protein, on trophoblast differentiation using the BeWo cell-line model. The authors utilize thapsigargin as a way to induce ER stress and release of CRT. Overall, the experiments are well thought out and the manuscript is well written however, the following comments need to be addressed:

Major comments:

  1. Apart from Figures 1 & 2, all the other figures and data are missing any type of quantitative and statistical analysis. While the western blot images are very convincing, but details about experimental design, technical replicates and n numbers, along with densitometry analysis is crucial.
  2. Also, in the title and throughout the discussion the authors indicate CRT affects syncytialization. While bHCG levels (for which no quantification is available) are a good indicator for differentiation, data verifying syncytialization- like syncytin staining or even assessment of fusion index in BeWo cells will significantly add to the manuscript.

Minor comments:

  1. Results section 3.1: The number of patient samples used in this section is confusing as the text says n=6, but the table provides details for n=22/18. The figure legend actually provides the distinction that n=20/18 was used for immunohistochemistry and IRS analysis. For the western blot analysis, the n=6. Can the authors please specify this in the results section.
  2. Negative control/IgG images for the immunostaining would be good to include.
  3. Figure 1. The blots for CRT in PE sera show an inconsistent pattern: do the authors have an explanation for this? Is CRT elevated in all PE patients?
  4. Also, what was the CRT serum levels normalized to? A GAPDH/Actin like control is missing. If it was normalized to amount of total protein, then this information is missing in the methods section.
  5. The authors mention few times that in their previous study (Ref 16) they observed that ablation of CRT causes decrease in syncytialization and E cadherin localization to cell surface – implying a role in differentiation. How do the authors put in context their current results which show similar effects, but with an increase in extracellular CRT? Addressing this in the discussion will help the reader to better understand the importance of the study.
  6. Lines 382-389: This particular section is very confusing. What other mechanisms other than ER stress can cause CRT release? Also, do sera of women with normal pregnancies show increase in CRT? As this contradicts the results shown and referenced.
  7. Lines 392-395: Please refer to minor comment 5.
  8. The discussion provides a lot of information and needs to be edited a little to ensure flow of information in the right context to help the reader.

Author Response

Point-by-point response to the reviewers

Reviewer #1:

Comments:

This manuscript outlines the effect of extracellular calreticulin (CRT),  an ER resident chaperone protein, on trophoblast differentiation using the BeWo cell-line model. The authors utilize thapsigargin as a way to induce ER stress and release of CRT. Overall, the experiments are well thought out and the manuscript is well written however, the following comments need to be addressed:

Major comments:

    Apart from Figures 1 & 2, all the other figures and data are missing any type of quantitative and statistical analysis. While the western blot images are very convincing, but details about experimental design, technical replicates and n numbers, along with densitometry analysis is crucial.

    Also, in the title and throughout the discussion the authors indicate CRT affects syncytialization. While bHCG levels (for which no quantification is available) are a good indicator for differentiation, data verifying syncytialization- like syncytin staining or even assessment of fusion index in BeWo cells will significantly add to the manuscript.

> First, we thank the reviewer for the careful and critical reading of our manuscript and for her/his constructive comments. We performed additional experiments and densitometric analyses for quantification. The results are now shown in Figures 4a, 4b, and 5a.

              We also determined fusion indexes for verification of syncytialization. We thank the reviewer for providing us an opportunity to strengthen our manuscript. The new results are included in Figures 4c and 5b. We also added missing information in the materials and methods section (2.9 Immunocytochemistry and determination of fusion index).

Minor comments:

  1. Results section 3.1: The number of patient samples used in this section is confusing as the text says n=6, but the table provides details for n=22/18. The figure legend actually provides the distinction that n=20/18 was used for immunohistochemistry and IRS analysis. For the western blot analysis, the n=6. Can the authors please specify this in the results section.

> We sincerely apologize for causing the confusion. Serum and placenta samples (n = 6 for each group) that were used in the immunoblot analysis were obtained from the patients shown in Table 1. The IHC was performed as described, “n = 18 for preeclamptic women and n = 22 for women with normal pregnancy”. We corrected and added description on page 5, lines 206–207, and page 5, lines 213–215.

  1. Negative control/IgG images for the immunostaining would be good to include.

> We thank the reviewer for pointing this out. We added an IHC image with an isotype rabbit IgG in Figure 1e.

  1. Figure 1. The blots for CRT in PE sera show an inconsistent pattern: do the authors have an explanation for this? Is CRT elevated in all PE patients?

> The reviewer raised an interesting point. Not all PE patients had elevated serum CRT, however, among the tested patients, the serum CRT levels in PE patients were significantly higher than those in women with normal pregnancy. Currently, we do not have an answer for why CRT in PE sera showed an inconsistent band pattern in the immunoblot. It might be correlate with the severity of the ER stress in PE patients, but the current manuscript focuses on the detrimental effect of extracellular CRT, hence we think that further elucidation of this issue will be an important future challenge. In order to show each band intensity in the immunoblots in Figure 1b, we replaced the bar graph of the densitometric analysis with bar graphs with scatter plots (Figure 1d).

  1. Also, what was the CRT serum levels normalized to? A GAPDH/Actin like control is missing. If it was normalized to amount of total protein, then this information is missing in the methods section.

> We thank the reviewer for the critical comment. In out immunoblot analysis of serum CRT, we normalized the loading amounts to the amount of the total proteins (100 µg/well). This information as well as sample preparation is now included in the material and methods section (page 4, lines 167–170).

  1. The authors mention few times that in their previous study (Ref 16) they observed that ablation of CRT causes decrease in syncytialization and E cadherin localization to cell surface – implying a role in differentiation. How do the authors put in context their current results which show similar effects, but with an increase in extracellular CRT? Addressing this in the discussion will help the reader to better understand the importance of the study.

> We apologize for causing a confusion. We noticed that although the accumulation of CRT in cis-Golgi was observed by occurrence of extracellular CRT in the present study and CRT knockdown cells in our previous study, our immunoblots indicated that extracellular CRT resulted in an altered E-cadherin band pattern. Thus, now we think that the mechanism of dysfunctional syncytialization by extracellular CRT is different from that by genetic depletion of CRT. We thank the reviewer for providing us an opportunity for correcting our misunderstanding. We discussed this issue on page 13, lines 424–433.

  1. Lines 382-389: This particular section is very confusing. What other mechanisms other than ER stress can cause CRT release? Also, do sera of women with normal pregnancies show increase in CRT? As this contradicts the results shown and referenced.

> Again, we apologize for causing a confusion. We and others observed elevated serum CRT in pregnant women, and our current results demonstrate further elevation of serum CRT in preeclamptic women. In our cell-based assays, we observed subtle CRT release in BeWo cells under general culture conditions, but marked CRT release in BeWo cells under an ER stress condition. Given that ER stress is one of the risk factors of the development of preeclampsia, we thought that CRT might occur in the sera of pregnant women with unknown reasons, and ER stress would contribute to the additional increase of serum CRT in preeclamptic women. However, this is speculative and confusing as pointed out by the reviewer. In order to avoid confusion, we modified the section on page 12, line 414–page 13, line 419.

  1. Lines 392-395: Please refer to minor comment 5.

> Please see our response to the minor comment 5.

  1. The discussion provides a lot of information and needs to be edited a little to ensure flow of information in the right context to help the reader.

> We thank the reviewer for pointing this out. We thoroughly revised the discussion taking the above comments into concern.

Reviewer 2 Report

This is an interesting study addressing the role of the ER chaperone calreticulin (CRT) in trophoblast function and its potential involvement in the pathogenesis of preeclampsia. Using expression analyses and in vitro functional studies in BeWo choriocarcinoma cells, the authors report the following main findings: 1) Preeclampsia patients exhibit up-regulated circulating levels of CRT compared to normal pregnancies, 2) ER stress promotes the extracellular release of CRT and 3) extracellular CRT interferes with forskolin-induced BeWo syncytialization. In general, the paper is well structured, the methods are accurately described and the results could potentially contribute to improve our understanding of the pathogenesis of preeclampsia. However, there are some methodological flaws that lessen the overall quality of the study and should be addressed before acceptance:

  1. Immunoblot analyses of CRT on serum samples: How were serum samples processed for western blot? Did you use any methods to concentrate the samples (i.e., dialysis)? This is not described in the Materials and Methods section. Concentration of circulating factors in serum are usually low for detection by western blot, this might be the reason why the bands in Fig. 1b look so weak and diffuse. Also in this figure, the loading control used for normalization is not presented, so it is not clear how the quantification in Fig 1d is performed. I’d suggest to use ELISA instead of westerns for quantification of serum samples. Alternatively, running the western with a prior concentration step and inclusion of appropriate loading controls would enhance robustness of the data presented.

 2. Tg-induced ER stress in BeWo cells: In Fig. 2a, which was the housekeeping gene used for normalization of PCR data? This information is omitted from the Materials and Methods. Ideally, real-time qPCR should be used for quantification of transcript levels but otherwise the results should be normalized against a housekeeping gene so as to provide an accurate representation of relative changes in gene expression. In Fig. 1b, please include a graphical representation of eIF2α expression levels and the corresponding statistical analysis.

Author Response

Point-by-point response to the reviewers

Reviewer #2:

This is an interesting study addressing the role of the ER chaperone calreticulin (CRT) in trophoblast function and its potential involvement in the pathogenesis of preeclampsia. Using expression analyses and in vitro functional studies in BeWo choriocarcinoma cells, the authors report the following main findings: 1) Preeclampsia patients exhibit up-regulated circulating levels of CRT compared to normal pregnancies, 2) ER stress promotes the extracellular release of CRT and 3) extracellular CRT interferes with forskolin-induced BeWo syncytialization. In general, the paper is well structured, the methods are accurately described and the results could potentially contribute to improve our understanding of the pathogenesis of preeclampsia. However, there are some methodological flaws that lessen the overall quality of the study and should be addressed before acceptance:

    Immunoblot analyses of CRT on serum samples: How were serum samples processed for western blot? Did you use any methods to concentrate the samples (i.e., dialysis)? This is not described in the Materials and Methods section. Concentration of circulating factors in serum are usually low for detection by western blot, this might be the reason why the bands in Fig. 1b look so weak and diffuse. Also in this figure, the loading control used for normalization is not presented, so it is not clear how the quantification in Fig 1d is performed. I’d suggest to use ELISA instead of westerns for quantification of serum samples. Alternatively, running the western with a prior concentration step and inclusion of appropriate loading controls would enhance robustness of the data presented.

> We first thank the reviewer for her/his careful reading, and the helpful and constructive comments for our manuscript. We are pleased that the reviewer appreciated the importance of our findings.

              In out immunoblot analysis of serum CRT, we applied 100 µg of total proteins in each well. Prior to immunoblotting, we diluted each serum sample with tris-buffered saline (pH 7.6) to obtain the final protein concentration of 10 µg/µL, and 100 µg of total proteins were subjected to SDS-PAGE and immunoblotting. Please also see our response to the minor comment 4 of Reviewer 1. We included this information in the material and methods section (page 4, lines 167–170). In order to improve the weak or diffuse bands in Figure 1b, we replaced the immunoblot images of Figure 1b with long-exposed ones, which clearly demonstrates CRT immunoreactive bands. We thank the reviewer for pointing this out.

  1. Tg-induced ER stress in BeWo cells: In Fig. 2a, which was the housekeeping gene used for normalization of PCR data? This information is omitted from the Materials and Methods. Ideally, real-time qPCR should be used for quantification of transcript levels but otherwise the results should be normalized against a housekeeping gene so as to provide an accurate representation of relative changes in gene expression. In Fig. 1b, please include a graphical representation of eIF2α expression levels and the corresponding statistical analysis.

> We thank the reviewer for the critical comments. We added the result of GAPDH as a housekeeping gene in Figure 2a. We also added quantification of phosphorylated eIF2α in Figure 2b.

Round 2

Reviewer 1 Report

The authors did a good job at addressing all of my concerns and made the necessary changes. 

Author Response

Reviewer #1:

Comments:

The authors did a good job at addressing all of my concerns and made the necessary changes.:

> We sincerely appreciate the kind support of the reviewer. We mentioned the results of fusion indexes (Figures 4c and 5b) in the result section on page 9, lines 321–322 and on page 11, lines 355–356, which was missing in the previous manuscript.

Reviewer 2 Report

The authors have satisfactorily answered most concerns and present an improved version of the manuscript. However, there are still minor concerns regarding data presentation and quantification that need to be addressed. In particular, the immunoblots for serum samples in Figure 1 still lack presentation of the loading controls (which the authors state is GAPDH), so it is difficult to assess differences in relative expression between the different patient groups. It is also not clear how the quantifications in Fig. 1c-d were performed: Y axis states "protein expression (relative to control)", is this fold-change respect to the control group? Band density should be normalized to GAPDH to calculate a relative expression for each sample, then these can be averaged and presented as fold-change relative to the control group.

Author Response

Reviewer #2:

Comments:

The authors have satisfactorily answered most concerns and present an improved version of the manuscript. However, there are still minor concerns regarding data presentation and quantification that need to be addressed. In particular, the immunoblots for serum samples in Figure 1 still lack presentation of the loading controls (which the authors state is GAPDH), so it is difficult to assess differences in relative expression between the different patient groups. It is also not clear how the quantifications in Fig. 1c-d were performed: Y axis states "protein expression (relative to control)", is this fold-change respect to the control group? Band density should be normalized to GAPDH to calculate a relative expression for each sample, then these can be averaged and presented as fold-change relative to the control group.:

> We thank the reviewer for the critical reading of our manuscript. In Figures 1c and 1d, the data are presented as fold-changes to the normal pregnancy group. We revised the Y axes as “Placental CRT protein expression (relative to “Normal pregnancy”)” and “Serum CRT protein level (relative to “Normal pregnancy”)”. For the immunoblot analysis of serum samples, we normalized band intensities to the amount of total proteins, i.e., 100 µg of total proteins were applied to each well. Thus, we think that loading amounts have been already normalized. For clarification, we mentioned this issue in the figure legend of Figure 1 (page 6, lines 232–233). Please also see the methods section on page 4, lines 167–170, which were added in response to both reviewers at the previous revision stage. We included protein concentrations of serum samples in newly prepared Supplemental Table S1.

              Please note that we mentioned the results of fusion indexes (Figures 4c and 5b) in the result section on page 9, lines 321–322 and on page 11, lines 355–356, which was missing in the previous manuscript.